# Unified Single Transformer for Multimodal Video Understanding and Generation

## Abstract

With the advancement of language models, unified multimodal understanding and generation have made significant strides, with model architectures evolving from separated components to unified single-model frameworks. This paper explores an efficient training paradigm to build a single transformer for unified multimodal understanding and generation. Specifically, we propose a multimodal warmup strategy utilizing prior knowledge to extend capabilities. To address cross-modal compatibility challenges, we introduce feature pre-scaling and multimodal AdaLN techniques. Integrating the proposed technologies, we present the HaploOmni, a new single multimodal transformer. With limited training costs, HaploOmni achieves competitive performance across image and video understanding and generation benchmarks over advanced unified models. All codes will be made public.

## 1 Introduction

In recent years, large-scale language models (LLMs) (Dubey et al., 2024; Yang et al., 2024a; Achiam et al., 2023) have exhibited remarkable capabilities across diverse domains, prompting researchers to extensively investigate their potential applications in multimodal contexts. There is an increasing focus on developing unified approaches that simultaneously address both multimodal understanding and generation capabilities. The former research can be categorized into three phases in terms of implementation architecture, progressing from segregated to unified frameworks.

In the first phase, tool-based methods like InstructGPT (Ouyang et al., 2022) and HuggingGPT (Shen et al., 2024) employ LLMs to allocate task-specific tools. While these methods offer simplicity and ease of use, their reliance on text-tool interactions limits their flexibility and controllability. In the second phase, methodologies incorporate separate encoders and decoders in conjunction with LLMs, exemplified by Seed (Ge et al., 2024), Emu-2 (Sun et al., 2024b), and VILA-U (Wu et al., 2024b), achieving multimodal input-output compatibility through feature interaction mechanisms. Although these approaches have achieved commendable results on general multimodal benchmarks, their segregated processes result in insufficient modal integration, constraining their capability to handle fine-grained understanding and generation tasks.

In the third phase, the latest approaches utilize a unified single-transformer framework. One subset, including Chameleon (Team, 2024) and Show-o (Xie et al., 2024), achieves model unification

| Method | Video Support | Single Transformer | Und. Data | Gen. Data | SEED | POPE | MVBench | VBench |
|---|---|---|---|---|---|---|---|---|
| SEED-X (Ge et al., 2024) | ✗ | ✗ | 152M | 152M | - | 84.2 | - | - |
| TokenFlow (Qu et al., 2024) | ✗ | ✗ | 10M | 60M | 68.7 | 86.8 | - | - |
| Janus-Pro (Chen et al., 2025) | ✗ | ✗ | 41M | 98M | 72.1 | 87.4 | - | - |
| Show-o (Xie et al., 2024) | ✗ | ✔ | 36M | 611M | - | 73.8 | - | - |
| ViLA-U (Wu et al., 2024b) | ✔ | ✗ | 7M | 16M | 59.0 | 85.8 | 38.9 | 73.4 |
| HaploOmni (ours) | ✔ | ✔ | **4M** | **3M** | **74.6** | **88.3** | **52.9** | **78.1** |

Table 1: Characteristics comparison with some other unified models. Video support means that the models can process video inputs and generate videos. "Und. Data" and "Gen. Data" indicate the number of training data for understanding and generation tasks, respectively.

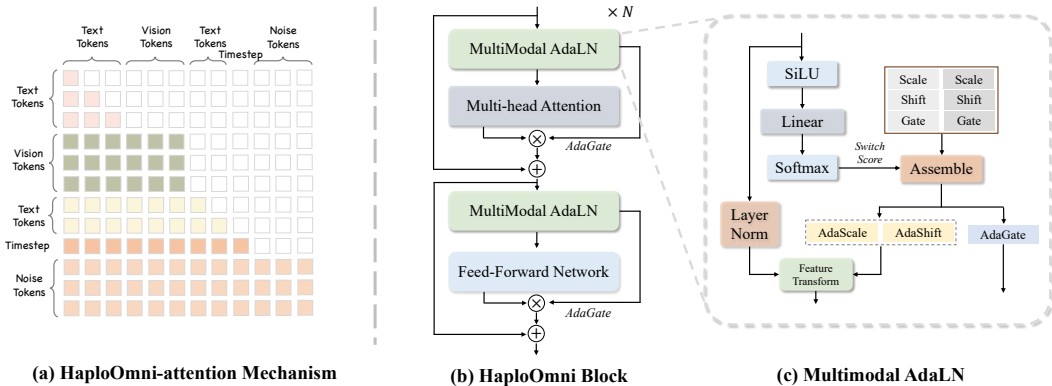

(a) HaploOmni-attention Mechanism     (b) HaploOmni Block     (c) Multimodal AdaLN

Figure 1: Illustration of our HaploOmni-attention mechanism and HaploOmni Block. We implement a hybrid masking strategy that applies causal attention to text features and timestep tokens (a vector $\in \mathbb{R}^{1 \times d}$) while adopting bidirectional attention for processing visual signals and latent noise. Drawing from the standard transformer module, we develop the HaploOmni block through the implementation of multimodal AdaLN.

through image discretization tokens. Another subset, exemplified by Transfusion (Zhou et al., 2024), employs hybrid text autoregressive and image diffusion modeling processes for unification. Compared to the encoder-decoder methods, these single-transformer methods are more streamlined and enable cross-modal early-fusion and late-fusion, thereby enhancing fine-grained multimodal representation capabilities (Zhou et al., 2024). However, existing methods adopt from-scratch training approaches. Due to the absence of prior knowledge, their overall performance falls short of encoder-decoder methods while incurring substantial training costs. Consequently, this paper explores a new perspective: *efficiently constructing a single multimodal transformer by leveraging knowledge from specialized models to achieve high-performance unified multimodal understanding and generation.*

To achieve it, we propose a new training paradigm for single multimodal transformers. Considering that the natural language possesses more abstract and higher-level semantic representations compared to natural images (Liu et al., 2024c), we propose a multimodal warmup process that depth-wise partitions a transformer decoder into three components: visual encoding, text encoding-decoding, and visual decoding. These components are initialized using corresponding prior models and subsequently fine-tuned independently to accommodate identity mapping across other modalities. Following the warmup phase, the model undergoes unified training for multimodal understanding and generation in an end-to-end manner. Furthermore, we find that different modalities exhibit varying preferences for feature scaling, significantly impacting training effectiveness and stability. Inspired by the diffusion transformer, we propose feature pre-scaling strategies and Multimodal AdaLN. The former pre-establishes initial feature transformation scales for different modalities based on statistical information, while the latter enables the model to autonomously select normalization parameters for various inputs.

With the proposed techniques, we present the **HaploOmni**, a cost-efficient yet high-performance single transformer for multimodal understanding and generation. As demonstrated in table 1, we evaluate our method on image and video multimodal understanding and generation benchmarks. Compared with previous models, our HaploOmni achieves superior performance across multiple image understanding datasets, including SEEDBench (Li et al., 2023a) and POPE (Li et al., 2023c). Additionally, it significantly outperforms unified video-text model, VILA-U, in both MVBench (Li et al., 2024b) video understanding and VBench (Huang et al., 2024) generation benchmarks.

## 2 RELATED WORK

**Text-to-video generation models** aim to automatically produce visually and logically consistent videos based on textual descriptions of scenes, objects, and actions. Most text-to-video models (Ho et al., 2022; Hong et al., 2022; He et al., 2022; Chen et al., 2023a) are built on latent diffusion models with a U-Net architecture. The field achieved a significant milestone with the introduction of diffusion transformers (Peebles & Xie, 2023), as demonstrated by the impressive Sora (openai, 2024). Following this breakthrough, the majority of studies have adopted diffusion transformers to

develop open-source text-to-video models. For example, CogVideoX (Yang et al., 2024c) introduces an expert transformer to improve the fusion of visual and textual modalities.

**Unified multi-modal LLMs** are capable of performing both understanding and generation tasks within a single framework. Several efforts (Ge et al., 2024; Wu et al., 2024b; Wang et al., 2024; Xiao et al., 2025) have been made to unify vision understanding and generation with an additional diffusion-based image decoder (Podell et al., 2023). In contrast, discrete sequence modeling methods (Team (2024); Wang et al. (2024); Wu et al. (2024b); Xie et al. (2024); Wu et al. (2024a); Qu et al. (2024)) discretize visual features and train token-based autoregressive models on mixed image and text data. It is worth mentioning that TransFusion (Zhou et al., 2024) attempts to integrate diffusion and autoregressive approaches within a single transformer. Despite recent advances, current approaches are limited by their inability to effectively trade off between performance and training resources, let alone extending to video understanding and generation area. In this paper, we introduce a method for efficiently constructing a unified single transformer achieving comparable performance across both vision understanding and generation tasks, including video area.

## 3 METHOD

In this section, we begin by introducing the preliminaries, followed by a detailed elaboration of our unified single transformer (HaploOmni) and the novel training paradigm we propose. This approach leverages knowledge from specialized models to efficiently construct HaploOmni, enabling high-performance unified multimodal understanding and generation.

### 3.1 PRELIMINARIES

**Multimodal LLMs.** Given a visual signal (image/video) and a series of corresponding text requests, a common approach for answer generation is to use a multimodal large language model (Liu et al., 2024c; Yang et al., 2024a; Chen et al., 2024b), which typically integrates a vision encoder and a language model. Generally, the raw visual input is transformed into a discrete or continuous feature space, which is then combined with text embeddings generated by a linguistic tokenizer. An auto-regressive LLM then processes the mixed multimodal sequence $\{x_t\}_{t=1}^{T-1}$ to predict the next tokens by modeling the conditional probability:

$$P\left(x_1, x_2, \cdots, x_T\right) = \prod_{t=1}^{T} P\left(x_t \mid x_1, x_2, \cdots, x_{t-1}\right).$$ (1)

Then, the $\mathcal{L}_{NTP}$ is defined using cross-entropy and the conditional probability described above, utilized to optimize the LLM during the training phase.

**Diffusion Transformer.** Diffusion models, such as the denoising diffusion probabilistic model (DDPM), generate data by progressively transforming noise into a target distribution over a series of timesteps. The Diffusion Transformer (DiT) integrates the transformer architecture into this generative process, enabling it to learn the reversal of the incremental noise-adding procedure in the forward process. At each timestep $t$, the model estimates the noise $\epsilon_t$ added to the data at the previous timestep. The objective function for training the Diffusion Transformer can be written as:

$$\mathcal{L}_{diff} = \mathbb{E}\left[\|\epsilon_t - \hat{\epsilon}_\theta(\mathbf{x}_t, t)\|^2\right]$$ (2)

where $\mathbf{x}_t$ is the noisy data at timestep $t$, $\mathbf{x}_0$ is the raw image or video data, and $\hat{\epsilon}_\theta(\cdot)$ is the model's estimate of the noise at each timestep, parameterized by the network $\theta$.

### 3.2 MODEL DESIGN

Overall, to streamline the training of the unified single transformer for multimodal understanding and generation, we first partition it into three components: pre-decoder, base-decoder, and post-decoder. All parts consist of multiple HaploOmni Blocks as shown in Fig. 1. Following this, two connector modules are employed to integrate the above three components into a complete transformer decoder. In contrast to previous decoupled paradigms (Ge et al., 2024; Sun et al., 2024b; Wu et al., 2024a), our unified architecture processes both visual and textual inputs together, eliminating

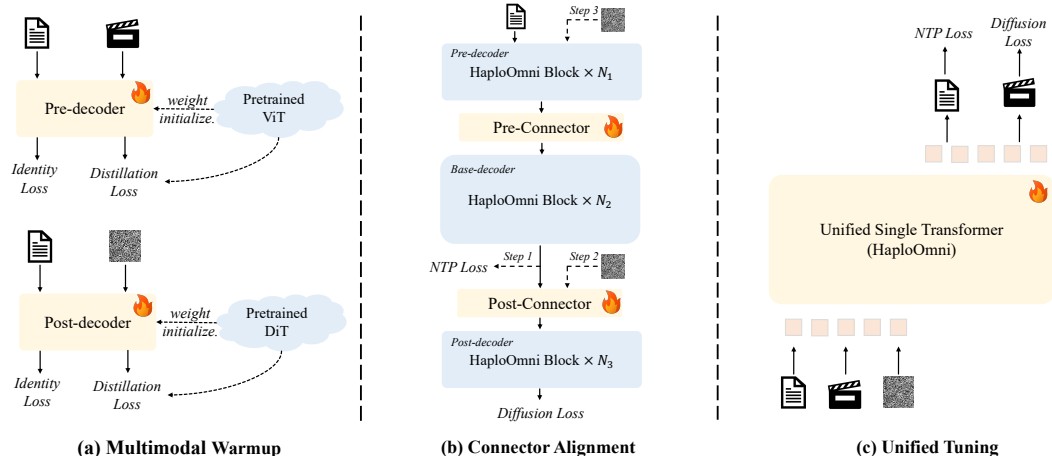

Figure 2: The progressive training stages of our HaploOmni, including multimodal warmup, connector alignment and unified tuning.

the need for a separate vision encoder, and enabling direct end-to-end visual generation conditioned on multimodal instructions. Additionally, we develop a three-stage training strategy to significantly reduce the training resources required. The following section provides an in-depth explanation of the specific modules within our HaploOmni.

**HaploOmni Block.** First, in light of the distinct characteristics of visual and linguistic modalities, we adopt a HaploOmni-attention mechanism with an adaptive mask strategy as shown in Fig. 1 (a) to improve multimodal representation capacity following previous methods (Xie et al., 2024; Zhou et al., 2024; Xiao et al., 2024). Inspired by the expert adaptive LayerNorm (AdaLN) introduced by CogVideoX to facilitate the fusion of different modalities by separately normalizing the condition and noise embeddings, we develop a multimodal AdaLN as shown in Fig. 1(c). Considering that AdaLN breaks the internal coherence required for constructing a one-transformer model, we introduce a dynamic strategy for input-aware normalization. Specifically, we compute the state matrix $S \in \mathbb{R}^{2\times3}$ offline to store two sets of scale, shift, and gate parameters as $\mathrm{SiLU}(\theta)W_{\mathrm{Ada}}^\top$, where SiLU, $\theta$, and $W_{\mathrm{Ada}}$ are the activation function, frozen time embedding, and a learnable matrix, respectively. Based on the input feature $h_i$ of the $i$-th token in the sequence, we compute two switch score sets used to perform a weighted summation over the discrete state matrix. The resulting AdaScale, AdaShift, and AdaGate parameters are then applied in the following feature transformation. The detailed operation is shown in algorithm 1. Leveraging the Multimodal AdaLN, we develop a HaploOmni block which is used to construct the complete model. The block is derived from the standard transformer structure, which includes two normalization layers, a feed-forward network, and an attention layer, with its execution order and residual method adhering to the original design, as depicted in Fig. 1 (b).

**Pre & Post Connector.** Integrating specific decoders leads to discrepancies in the feature space across different modalities, which poses challenges for joint training and modality fusion. To alleviate this, we introduce a novel connector module with multimodal LN to align the modalities within a unified feature space. Specifically, given a multimodal sequence $X$ with the length of $L$ concatenated by $\{x_1, x_2\}$ where $\{x_1, x_2\}$ indicates the condition tokens and latent noise tokens respectively, we utilize a set of LayerNorm with learnable transition matrices $W'$ to process the sequence as follows:

$$\widetilde{X} = \mathrm{SiLU}(\mathrm{LayerNorm}(X))W' \qquad (3)$$

---

**Algorithm 1** Multimodal AdaLN

**Input:**
$\quad h_i \in \mathbb{R}^{1\times d}$         ▷ *Input feature*
$\quad W_{\mathrm{MAL}} \in \mathbb{R}^{d\times 2}$     ▷ *Input learnable matrix*
$\quad S \in \mathbb{R}^{3\times 2}$         ▷ *State matrix*

**Forward:**
$\quad \overline{h_i} \leftarrow \dfrac{h_i}{1+exp(-h_i)} W_{\mathrm{MAL}}^\top$
$\quad$ **Set** $\delta \in \mathbb{R}^{1\times 2}$
$\quad \delta_k \leftarrow \dfrac{exp(\overline{h_i^k})}{\sum_{j=1}^{2} exp(\overline{h_i^j})}$   ▷ *Switch Score*
$\quad [\mathrm{AdaScale, AdaShift, AdaGate}] \leftarrow \delta S^\top$
$\quad$ **Do:**         ▷ *Feature Transform*
$\quad \widetilde{h_i} \leftarrow (\mathrm{AdaScale}+1) \times LN(h_i) + \mathrm{AdaShift}$

**Output:** $\widetilde{h_i}$, AdaGate

---

Then, we obtain the corresponding switch scores $P^{\text{score}} \in \mathbb{R}^{L \times 2}$ through an indicator layer consisting of a SiLU function, a learnable matrix $W_{\text{SN}}$, and a Softmax function ($\sigma$), which can be formulated as:

$$P^{\text{score}} = \sigma(W_{\text{SN}}(\text{SiLU}(X))) \tag{4}$$

With a characteristic function $\mathbb{I}$, the score is multiplied by the input $\widetilde{X}$ to obtain the well-aligned feature $\{X'_i\}_{i=1}^{L}$:

$$X'_i = \mathbb{I}_0(P^{\text{score}})\widetilde{X}_i + \mathbb{I}_1(P^{\text{score}})X_i \tag{5}$$

**Feature Pre-scaling.** Although the model can ultimately be optimized through the connector we designed, the optimization process is relatively slow. We observe that aligning features across modalities gives rise to considerable amplitude inconsistencies, with the amplitudes of noise tokens often being about 10 times larger than those of the visual features distilled by a prior ViT. This disparity intensifies feature-space distribution differences, complicating the training process. Additionally, in our paradigm, small perturbations near extreme points, stemming from the pre-trained model, lead to diminished gradient amplitudes, which slow parameter updates. Therefore, we introduce a feature pre-scaling mechanism into the Pre and Post-decoder, significantly simplifying training and accelerating model convergence.

**Inference Mode.** In the inference stage, our model uses a unified transformer to execute multimodal understanding and generation tasks seamlessly. For the understanding task, given a visual signal such as an image or video and a corresponding text query, the visual input is first converted into a sequence via a patchification layer, while the text is tokenized into a sequence. The concatenated multimodal sequences are then fed into the transformer and output with the corresponding response. For the generation task, we combine condition tokens (typically text embeddings), a timestep token (a vector $\in \mathbb{R}^{1 \times d}$ calculated in the same way as the time embedding in the classic diffusion model) and random noise tokens into a multimodal sequence, process it iteratively through a unified transformer according to the DDIM (Song et al., 2020) schedule, and decode the resulting latent representation into the final image or video using a VAE decoder (Yang et al., 2024c).

### 3.3 TRAINING PROCEDURE

HaploOmni is initially partitioned into three components: pre-decoder, base-decoder, and post-decoder. We then train these components in three distinct stages: Multimodal Warmup, Connector Alignment, and Unified Tuning, as shown in Fig. 2.

**Stage 1: Multimodal Warmup.** The three sub-decoders are first initialized using the corresponding prior models and subsequently fine-tuned independently to accommodate identity mapping across other modalities. At this stage, we only train pre-decoder and post-decoder to ensure they conform to the auto-regressive paradigm without altering the original model's learnable parameters. This adjustment enables compatibility with the LLM reasoning framework, including KV-Cache, temperature setting, and top-p truncation. For the pre-decoder, a mixed sequence of text and image tokens is used as input and we leverage the HaploOmni-attention mechanism for multimodal interaction. Two losses are applied during training: Identity Loss for linguistic modality and distillation loss to preserve visual knowledge while learning new text-based knowledge. On the other hand, we train the denoising capability of the post-decoder with randomly noisy video as input, conditioned by the corresponding text description, while applying both distillation loss and identity loss. Given the input signal, $x^{\text{image}}$, $x^{\text{text}}$ for pre-decoder while $x^{\text{text}}$ and $x^{\text{latent}}$ for post-decoder, the corresponding outputs are $y^{\text{image}}$, $y^{\text{text}}_{\text{pre}}$, $y^{\text{text}}_{\text{post}}$ and $y^{\text{latent}}$. The CLIP-ViT outputs $y^{\text{image}}_{\text{vit}}$ and CogVideoX outputs $y^{\text{latent}}_{\text{cog}}$, the objective functions of pre and post-decoder in this training stage are formulated as:

$$\mathcal{L}_{pre} = \lambda_1^{\text{pre}} ||x^{\text{text}} - y^{\text{text}}_{\text{pre}}||^2 + \lambda_2^{\text{pre}} ||x^{\text{image}} - y^{\text{image}}_{\text{vit}}||^2 + \lambda_3^{\text{pre}} \frac{x^{\text{image}} y^{\text{image}}_{\text{vit}}}{||x^{\text{image}}|| \cdot ||y^{\text{image}}_{\text{vit}}||}$$

$$\mathcal{L}_{post} = \lambda_1^{\text{post}} ||x^{\text{text}} - y^{\text{text}}_{\text{post}}||^2 + \lambda_2^{\text{post}} ||x^{\text{latent}} - y^{\text{latent}}_{\text{cog}}||^2 + \lambda_3^{\text{post}} \frac{x^{\text{latent}} y^{\text{latent}}_{\text{cog}}}{||x^{\text{latent}}|| \cdot ||y^{\text{latent}}_{\text{cog}}||} \tag{6}$$

| Type | Model | Size | SEED↑ | POPE↑ | AI2D↑ | RWQA↑ | MMMU↑ | MMB(test)↑ | MMStar↑ | VQAv2↑ | GQA↑ |
|------|-------|------|-------|-------|-------|-------|-------|-----------|---------|--------|------|
| *Und.* | Qwen-VL-Chat (Bai et al., 2023) | 7B | 58.2 | - | 45.9 | 49.3 | 35.9 | 60.6 | 37.5 | 78.2 | 57.5 |
| *Only* | InternVL-Chat (Chen et al., 2024b) | 7B | - | 86.4 | 54.8 | - | - | - | - | 79.3 | 62.9 |
| | ShareGPT4V (Chen et al., 2023b) | 7B | - | - | 58.0 | 54.9 | 37.2 | 68.8 | 33.0 | 80.6 | 63.3 |
| | LLaVA-1.6 (Liu et al., 2024b) | 7B | 64.7 | 86.5 | 66.6 | 57.8 | 35.1 | 67.4 | - | 81.8 | 64.2 |
| | LLaVA-OV (Li et al., 2024a) | 7B | 75.4 | - | 81.4 | 66.3 | 48.8 | 80.8 | 61.7 | - | - |
| | Fuyu-8B (Bavishi et al., 2023) | 8B | - | 74.1 | 64.5 | - | 27.9 | 10.7 | - | 74.2 | - |
| | EVE-7B (Diao et al., 2024) | 8B | 54.3 | 83.6 | - | - | - | 49.5 | 28.2 | 75.4 | 60.8 |
| | Emu3-Chat (Wang et al., 2024) | 8B | 68.2 | 85.2 | 70.0 | 57.4 | 31.6 | 58.5 | - | 75.1 | 60.3 |
| *Und.* | NExT-GPT (Wu et al., 2023) | 13B | - | - | - | - | - | - | - | 66.7 | - |
| *and* | VILA-U (Wu et al., 2024b) | 8B | 59.0 | 85.8 | - | - | - | - | - | 79.4 | 60.8 |
| *Gen.* | Janus-Pro (Chen et al., 2025) | 8B | 72.1 | 87.4 | - | - | 41.0 | 79.2 | - | - | 62.0 |
| | Chameleon (Team, 2024) | 30B | - | - | - | - | - | 37.6 | - | 69.6 | - |
| | Show-o (Xie et al., 2024) | 1.3B | - | 73.8 | - | - | 25.1 | - | - | 59.3 | 48.7 |
| | TokenFlow-XL (Qu et al., 2024) | 13B | 68.7 | 86.8 | 66.7 | 53.7 | 38.7 | 68.9 | - | 77.9 | 62.7 |
| | **HaploOmni (ours)** | 9B | **74.0** | **89.6** | **78.7** | **63.5** | **46.1** | 78.2 | **57.8** | 75.6 | 60.8 |

Table 2: Comparison with state-of-the-arts on image understanding benchmarks. "Und." and "Gen." denote "understanding" and "generation", respectively. Models below the dotted line are the single-transformer methods. Bold indicates the best result, while underlined marks the second-best.

**Stage 2: Connector Alignment.** This stage aims to optimize the model training cycle across three progressive steps. In the first step, the pre-connector is trained on multimodal understanding tasks with $\mathcal{L}_{NTP}$. In the second step, we train the post-connector, equipping the post-decoder to handle video and image denoising based on semantic features from the base LLM with diffusion loss, which is formulated as $\mathcal{L}_{diff}$. Finally, we train both the pre-connector and post-decoder to allow the entire model to process visual, text, and latent noise features directly in an end-to-end manner.

| Model | Size | EgoSchema↑ | MVBench↑ |
|-------|------|-----------|----------|
| *Und. only* | | | |
| LLaMA-VID (Li et al., 2025) | 7B | 38.5 | 41.9 |
| Video-LLaVA (Lin et al., 2023) | 7B | 38.4 | 41.0 |
| VideoChat2 (Li et al., 2024b) | 7B | 42.2 | 51.1 |
| *Und. and Gen.* | | | |
| Video-LaVIT (Jin et al., 2024) | 7B | 37.3 | - |
| VILA-U (Wu et al., 2024b) | 8B | 33.4 | 38.9 |
| **HaploOmni (ours)** | 9B | 47.1 | 52.9 |

Table 3: Comparison on Video understanding benchmarks. "Und." and "Gen." denote "understanding" and "generation", respectively.

**Stage 3: Unified Tuning.** At this stage, we integrate the three decoders into a unified single transformer (HaploOmni). The entire model is fine-tuned using a combination of understanding and generation datasets. Inputs across all modalities are uniformly processed through HaploOmni, which then generates the corresponding output. In this stage, we leverage both $\mathcal{L}_{NTP}$ and $\mathcal{L}_{diff}$ to optimize HaploOmni.

## 4 EXPERIMENTS

We conduct extensive experiments to evaluate the effectiveness of our HaploOmni and compare it to the widely adopted large language model approaches on multimodal understanding and generation tasks under a fair evaluation setting. More ablation results can be shown in the Appendix.

### 4.1 IMPLEMENTATION DETAILS

The base-decoder of our HaploOmni is based on Qwen2.5-7B (Yang et al., 2024b). During the distillation stage, we employ CLIP-ViT-L and CogVideoX-2B as the teacher models for the pre-decoder and post-decoder, respectively, with the decoders comprising 24 and 30 layers ($N_1$ and $N_2$). Due to the limited space, more implementation details are shown in Appendix.

**Datasets.** We classify image-text data pairs for multimodal understanding into three types consisting of 1.7M image caption data (Chen et al., 2023b; Liu et al., 2024c), 1.2M single-image instruction data (Liu et al., 2024a; Li et al., 2024a) and 1.1M interleaved multi-image and video datasets (Zhu et al., 2023; Zhang et al., 2024; Li et al., 2024a). For the vision generation task, we curated 2M Jour-

| Type | Model | Subject Consistency↑ | Scene↑ | Dynamic Degree↑ | Motion Smoothness↑ | Background Consistency↑ |
|---|---|---|---|---|---|---|
| *Gen. Only* | OpenSora-V1.1 (Lin et al., 2024) | 96.8 | 27.2 | 47.7 | 98.3 | 97.6 |
| | AnimateDiff-V2 (Guo et al., 2023) | 95.3 | 50.2 | 40.8 | 97.8 | 97.7 |
| | Pika (Pika, 2023) | 96.9 | 49.8 | 47.5 | 99.5 | 97.4 |
| | VideoCrafter-2.0 (Chen et al., 2023a) | 96.9 | 55.3 | 42.5 | 97.7 | 98.2 |
| | CogVideoX-5B (Yang et al., 2024c) | 96.2 | 53.2 | 71.0 | 96.9 | 96.5 |
| | Kling (Kling, 2024) | 98.3 | 50.9 | 46.9 | 99.4 | 97.6 |
| | Gen-3 (Runway, 2024) | 67.1 | 54.6 | 60.1 | 99.2 | 96.6 |
| | Emu3-gen (Wang et al., 2024) | 95.3 | 37.1 | 79.3 | 98.9 | 97.7 |
| *Und. and Gen.* | VILA-U (Wu et al., 2024b) | 87.0 | 31.8 | 58.7 | 95.3 | 94.4 |
| | **HaploOmni (ours)** | **96.4** | **34.6** | **65.3** | **96.8** | **97.6** |

Table 4: Comparison with state-of-the-arts on video generation benchmark, VBench (Huang et al., 2024). "Und." and "Gen." denote "understanding" and "generation", respectively. Bold indicates the best result, while underlined marks the second-best.

| Pre-Scaling | MM Warmup | MM AdaLN | MMMU | MVBench | VBench |
|---|---|---|---|---|---|
| | | | 34.4 | 46.3 | 68.1 |
| | ✔ | | 40.1 | 48.3 | 72.7 |
| | ✔ | ✔ | 42.7 | 50.2 | 74.4 |
| ✔ | ✔ | ✔ | 46.1 | 52.9 | 78.1 |

Table 5: "MM" denotes Multimodal. The effectiveness of our proposed modules and algorithms across three tasks: vision understanding (image/video) and video generation, as on MMMU (Yue et al., 2024), MVBench (Li et al., 2024b) and VBench (Huang et al., 2024) benchmarks, respectively.

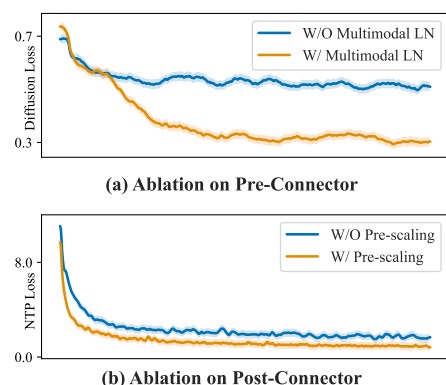

(a) Ablation on Pre-Connector

(b) Ablation on Post-Connector

Figure 3: Loss curve comparison of different settings. The x-axis is the training step.

neyDB (Sun et al., 2024a) image-text pairs and approximately 1M video generation data, including 374K WebVid (Bain et al., 2021), 626K in-house data. More details are shown in Appendix.

## 4.2 MAIN RESULTS

**Vision Understanding.** We provide a comparative analysis of state-of-the-art models on vision understanding task across various benchmarks as depicted in Tab. 2 and Tab. 3 involving image and video, respectively. As depicted in Tab. 2, our HaploOmni, as a unified multimodal model outperforms existing methods on most evaluation metrics. HaploOmni achieves state-of-the-art results among unified models on most benchmarks, with notable scores such as 74.8 on SEED and 87.9 on POPE, surpassing prior approaches like Janus and VILA-U. Additionally, HaploOmni demonstrates competitive performance compared to understanding-only models, achieving scores of 76.6 on AI2D and 60.8 on RWQA, outperforming Emu3-chat by +6.6% and +3.4%, respectively. Furthermore, the comparison results in Tab. 3 highlight HaploOmni's impressive video understanding capabilities. Specifically, HaploOmni achieves 47.1 on EgoSchema and 52.9 on MVBench, surpassing Video-LaVIT with 37.3 on EgoSchema, and VILA-U with 38.9 on MVBench.

**Video Generation.** We compare the performance of our proposed HaploOmni with state-of-the-art video generation models on the VBench benchmark as shown in Tab. 4. Following previous works (Wang et al., 2024; Yang et al., 2024c) on video generation, we selected some aspects that can reflect the quality of the generated video, like dynamic degree, subject consistency, and motion smoothness. HaploOmni as a unified model, exhibits strong performance across most evaluated aspects. Specifically, we achieve a Scene Consistency score of 96.4, outperforming other multimodal

| Type | MMMU-val | MMStar | AI2D |
|---|---|---|---|
| Standard | 34.4 | 68.1 | 72.3 |
| HaploOmni-Block | 39.7 | 73.4 | 76.6 |

Table 6: Effectiveness of HaploOmni Block. A standard block refers to a commonly used block architecture in large language models with pretrained weights of LLaMA-3.

| | Chameleon | Janus | HaploOmni |
|---|---|---|---|
| Support Video | ✗ | ✗ | ✔ |
| GPUs Hours | 856481 | 21504 | 5792 |

Table 8: Comparison of training GPUs hours among some unified multi-modal large language models.

| LLM Type | Size | MMMU-val | SEED | POPE |
|---|---|---|---|---|
| LLaMA-2 | 7B | 34.7 | 66.1 | 86.2 |
| LLaMA-3 | 8B | 39.7 | 74.8 | 87.9 |
| Qwen-2.5 | 7B | 46.1 | 74.0 | 89.6 |

Table 7: Performance comparison with different LLM backbones.

| | ImageNet Acc. | VBench Overall. |
|---|---|---|
| CLIP-ViT-L | 79.2 | ✗ |
| Pre-decoder | 79.0 | ✗ |
| CogVideox-2B | ✗ | 77.3 |
| Post-decoder | ✗ | 78.3 |

Table 9: Performance Comparison after Multimodal Warmup training stage.

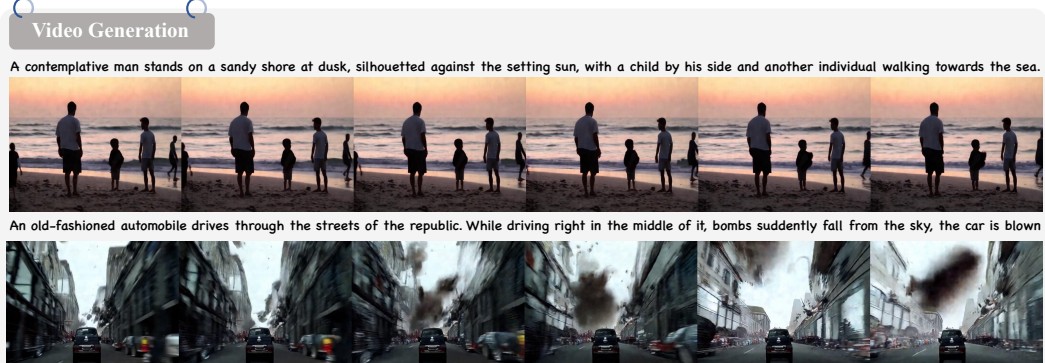

Figure 4: Qualitative results of HaploOmni. The resolution of all the generated videos is 480 ×720.

models like VILA-U (87.0) while remaining competitive with pure generative models such as Kling (98.3) and Pika (96.9).

### 4.3 ABLATION STUDY

We conduct various analysis experiments and present some visual results to illustrate the effectiveness of our method. As shown in Fig. 3, multimodal LN effectively reduces the difficulty of visual generation training, while feature pre-scaling accelerates the training process for multimodal understanding and improves loss convergence. They are both beneficial for performance improvement as illustrated in Tab. 5. We ablate various strategies by generating a cute cat as illustrated in Fig. 7. Noise increases when multimodal AdaLN is absent. As shown in Tab. 6, our HaploOmni Block outperforms the standard version under a fair evaluation protocol, which demonstrates the effectiveness of our architectural design. Tab. 7 illustrates different performance across three common LLM backbones. Moreover, we have evaluated our pre and post-decoder after the first training stage as shown in Tab. 9. The former achieves comparable performance with its teacher while the latter outperforms its teacher, which emphasizes the effectiveness of the Multimodal Warmup stage.

### 4.4 QUALITATIVE RESULTS

To better illustrate the capabilities of our HaploOmni, we provide examples of image understanding, video understanding, and video generation. As shown in Fig. 5, with the decoder-only architecture, the model can handle input images of varying resolutions and perceive the fine-grained information. Meanwhile, HaploOmni effectively displays the motion range of generated concepts, such as the butterfly in Fig. 6 and building fragments in Fig. 4. More qualitative results are shown in Fig. 8.

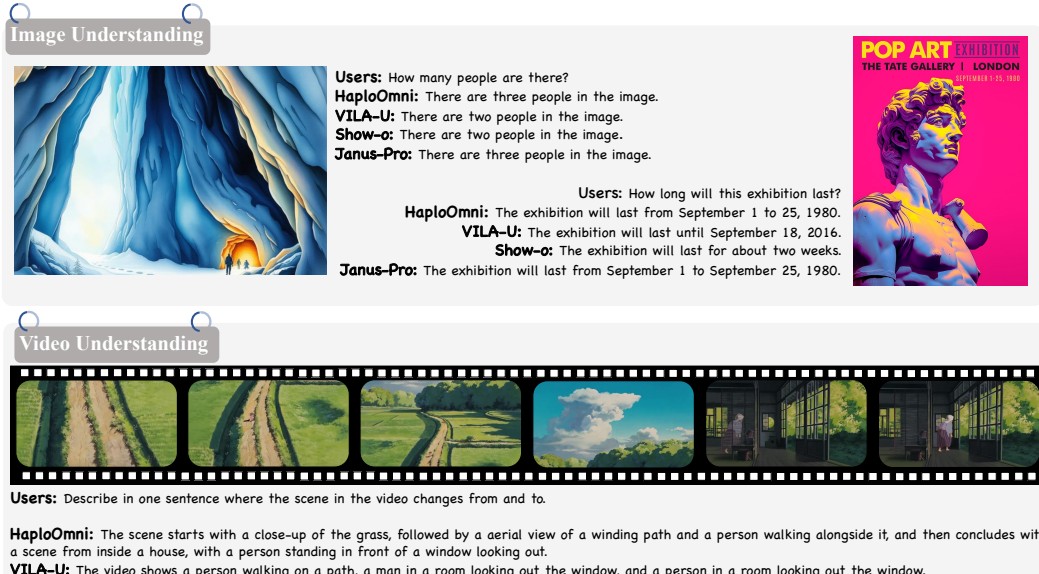

Figure 5: Performance comparison on image and video understanding capability.

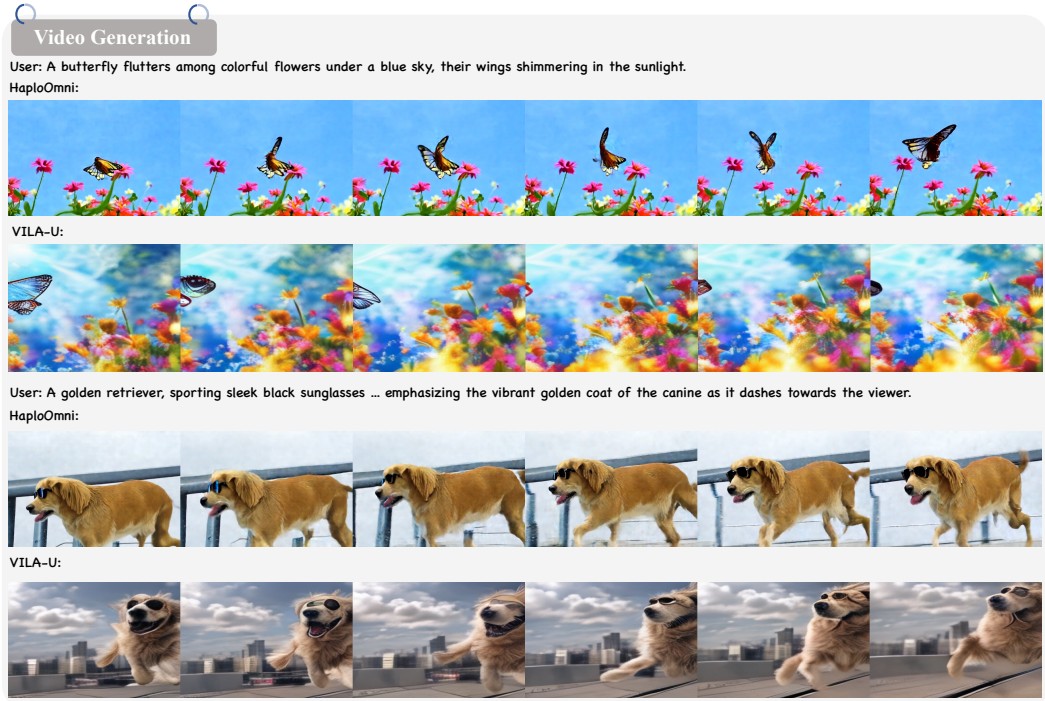

Figure 6: Performance comparison on video generation capabilities. The resolution of the generated video is 480 ×720.

## 5 CONCLUSION

This paper explores a new training paradigm for single multimodal transformers. By introducing a multimodal warmup strategy incorporating prior knowledge, we substantially reduce training complexity and computational costs. Furthermore, we propose the feature pre-scaling strategy and multimodal AdaLN to address cross-modal integration challenges. With these techniques, our proposed HaploOmni demonstrates high performance in both image and video understanding and generation, achieving state-of-the-art results across multiple benchmarks. Additionally, we believe our methodological approach can inspire future LLM-based research.

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

# APPENDIX

## A IMPLEMENTATION DETAILS

### A.1 DATASETS.

We classify image-text data pairs for multimodal understanding into three types: 1) image caption data, which include 1.2M ShareGPT4V-PT (Chen et al., 2023b) and 558K LLaVA pretraining data (Liu et al., 2024c); 2) single-image instruction data, comprising 665K LLaVA v1.5 (Liu et al., 2024a) and 0.5M public dataset (Li et al., 2024a); and 3) interleaved multi-image and video datasets, which consist of 0.6M CC3M (Zhu et al., 2023), LLaVA-Hound mixed data, and 0.5M video datasets (Zhang et al., 2024; Li et al., 2024a). Furthermore, we follow existing works (Wu et al., 2024a; Xie et al., 2024; Yang et al., 2025) to organize the above caption data into question-answering pairs. For the visual generation task, we curated 2M JourneyDB (Sun et al., 2024a) image-text pairs and approximately 1M video generation datasets, including 374K WebVid (Bain et al., 2021), 626K in-house data.

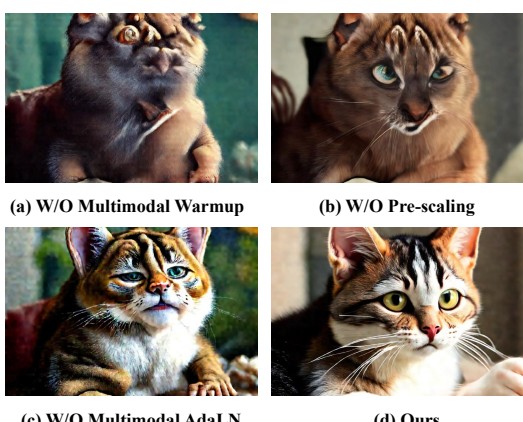

|  |  |
|---|---|
| **(a) W/O Multimodal Warmup** | **(b) W/O Pre-scaling** |
| **(c) W/O Multimodal AdaLN** | **(d) Ours** |

Figure 7: Visualization results for the ablation of different components. (d) indicates the final version of our model.

### A.2 METRICS.

In multimodal understanding, our model HaploOmni is evaluated on widely adopted image-based benchmarks. including GQA (Hudson & Manning, 2019), VQAv2 (Goyal et al., 2017), AI2D (Kembhavi et al., 2016), MMBench-EN-dev (MMB), MMMU (Yue et al., 2024), RealWorldQA, MM-Star (Chen et al., 2024a), POPE (Li et al., 2023c) and SEED-Bench-IMG (SEED) (Li et al., 2023b) as well as the video benchmarks, including MVbench (Li et al., 2024b) and EgoSchema (Mangalam et al., 2023). For generation tasks, we evaluate our model on VBench (Huang et al., 2024) , which involves various metrics such as dynamic degree, motion smoothness, and subject consistency.

### A.3 IMPLEMENTATION.

The base-decoder of our HaploOmni is based on Qwen2.5 (Yang et al., 2024b). During the distillation stage, we employ CLIP-ViT-L and CogVideoX-2B as the teacher models for the pre-decoder and post-decoder, respectively, with the decoders comprising 24 and 30 layers ($N_1$ and $N_2$). In the decoder warmup stage, the pre-decoder is trained with a learning rate of 1e-4 and a batch size of 256, while the post-decoder is trained using a learning rate of 2e-4 and a batch size of 32. In step 1 of the alignment stage, we align the pre-decoder and mid-decoder with a learning rate of 1e-5 and a batch size of 128, training only the pre-connector with a 2K-step warmup. In step 2, the pre-connector is warmed up for 10K iterations using JourneyDB data with a learning rate of 1e-4 and a batch size of 128, after which we relax the training for the post-decoder. In step 3, we train the pre-connector, post-connector, and post-decoder with the same settings, enabling end-to-end input-output of latent

| $\lambda_1^{\mathbf{pre}}$ | $\lambda_2^{\mathbf{pre}}$ | $\lambda_3^{\mathbf{pre}}$ | ImageNet Acc |
|------|------|------|------|
| 1.0 | 1.0 | 0.5 | 76.2 |
| 1.0 | 0.8 | 0.5 | 77.8 |
| 1.0 | 0.8 | 1.2 | 78.3 |
| 1.0 | 1.0 | 1.0 | 79.0 |

Table 10: Ablation about loss coefficient of pre-decoder in Multimodal Warmup training stage.

| $\lambda_1^{\mathbf{post}}$ | $\lambda_2^{\mathbf{post}}$ | $\lambda_3^{\mathbf{post}}$ | VBench (Overall) |
|------|------|------|------|
| 1.0 | 1.0 | 0.4 | 77.0 |
| 1.0 | 0.8 | 0.3 | 76.2 |
| 1.0 | 0.8 | 1.2 | 75.3 |
| 1.0 | 1.0 | 1.0 | 78.3 |

Table 11: Ablation about loss coefficient of post-decoder in Multimodal Warmup training stage.

| Type | MMMU | VBench |
|------|------|------|
| Causal | 45.7 | 58.6 |
| Omni | 46.1 | 78.1 |

Table 12: Ablation about different attention mechanisms. Omni indicates our HaploOmni-attention mechanism.

| Type | Method | User Study |
|------|------|------|
| Und. & Gen. | VILA-U | 14% |
| Gen. only | CogVideoX-2B | 42% |
| Und. & Gen. | HaploOmni | 44% |

Table 13: User studies about different types of models.

features. Finally, in the third stage, the HaploOmni is fine-tuned uniformly with mixed video and image generation, as well as multimodal understanding data with a learning rate of 2e-5 and a batch size of 32. Across all experiments, the AdamW optimizer is configured with betas (0.9, 0.999) and a momentum of 0.9 (Loshchilov & Hutter, 2017). By default, the number of multimodal AdaLN layers is set to 2.

### A.4 LIMITATION ANALYSIS

We focus on providing a framework that efficiently develops a unified single-transformer video model within a decoder-only paradigm, substantially narrowing the gap compared to previous unified approaches. Nevertheless, there remains considerable room for improving the model's runtime efficiency, which we plan to investigate in future work.

## B USE OF LARGE LANGUAGE MODELS

We used GPT-5 (OpenAI, 2025) exclusively for language polishing, such as grammar, clarity, and style. The model was not involved in generating ideas, designing experiments, or interpreting results. All technical content was independently written, verified, and approved by the authors.

## C ETHICS STATEMENT

This work mainly relies exclusively on publicly available, open-source datasets that have been widely used in prior academic research. All datasets are employed strictly for scholarly purposes and will not be used in any commercial applications.

## D REPRODUCIBILITY STATEMENT

To support reproducibility, we will release the project as open-source software. The model architecture is described in detail in Section 3, and Section 4 outlines the complete training pipeline, implementation details, and all hyperparameter settings to enable faithful replication.

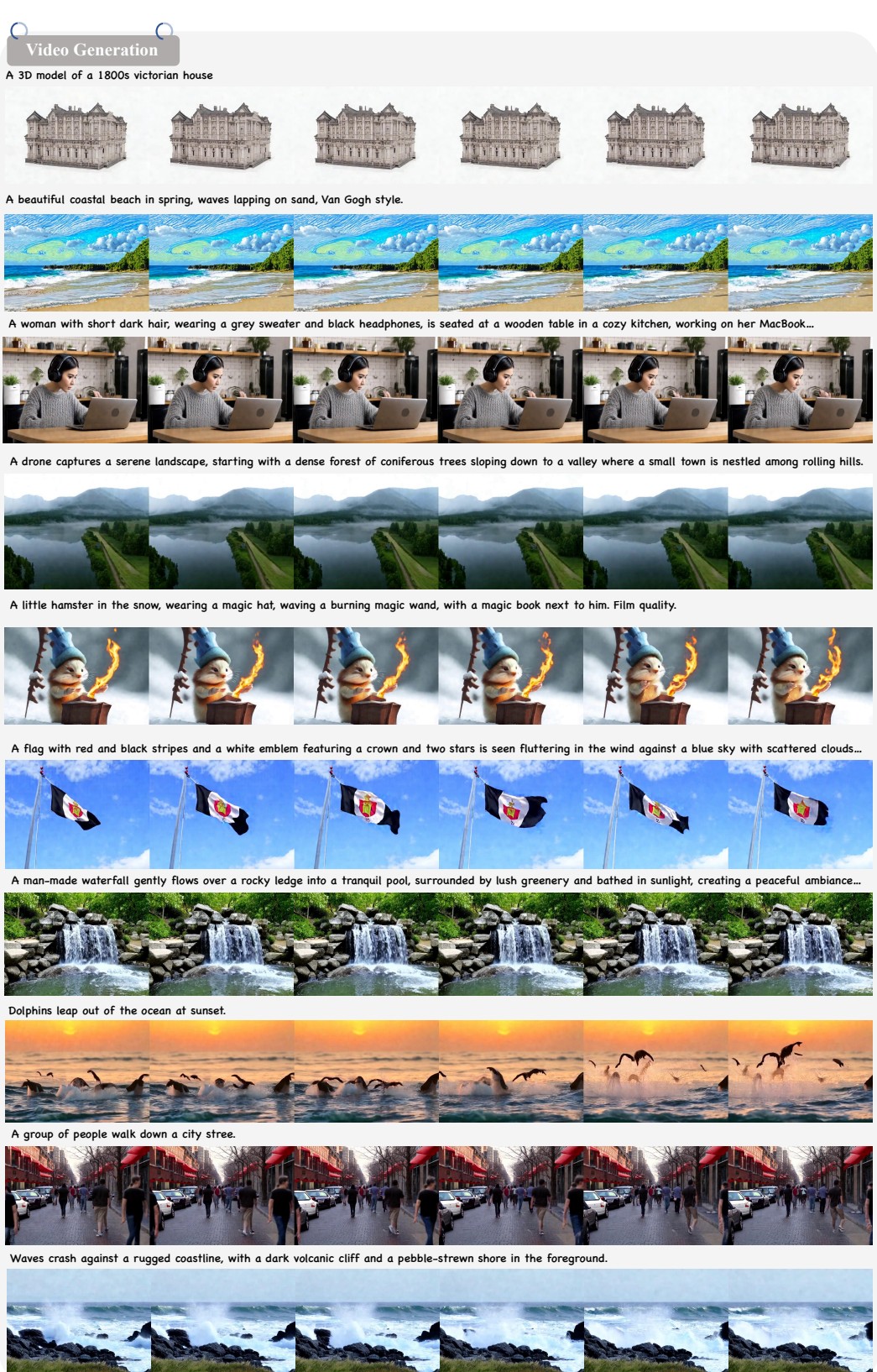

Figure 8: More qualitative results about video generation.

