# OpenReview forum: "Unified Single Transformer for Multimodal Video Understanding and Generation"
_ICLR.cc/2026/Conference — ICLR 2026 Conference Withdrawn Submission_

### Official Review · Reviewer_Ue3m · 2025-10-27

**Soundness:** 2
**Presentation:** 3
**Contribution:** 2
**Rating:** 4
**Confidence:** 4

**Summary:**

The paper proposes a single decoder-only transformer for multimodal understanding and generation (images and video). The approach warms up three sub-decoders (pre/base/post) with prior models, applies feature pre-scaling to normalize modality amplitudes, and introduces a Multimodal AdaLN with input-aware switches. These components are connected via pre/post connectors and then tuned end-to-end in three stages (warmup → connector alignment → unified training). Experiments report competitive results on image/video understanding and VBench generation with comparatively modest training time, and the authors plan to release code. However, the novelty feels incremental because Multimodal AdaLN and pre-scaling read as stabilization tweaks rather than a new principle, and their impact is not isolated by matched ablations. Reporting has rough edges with text–table inconsistencies and mixed comparisons between specialized and unified models, so the strength of the evidence is hard to gauge.

**Strengths:**

The paper lays out a practical path to a unified model that is straightforward to implement and scale. The staged warmup makes effective use of strong teachers to ease cold starts and stabilize early training. Feature pre-scaling and the proposed Multimodal AdaLN act as simple, broadly useful stabilizers for mixed-modality learning. The ablations show additive gains on image and video understanding (MMMU, MVBench) and on video generation (VBench). The GPU-hours table suggests the approach reaches competitive quality with a comparatively modest compute budget relative to several unified baselines.

**Weaknesses:**

1. **What is actually new is hard to see:** The model is presented as a single-transformer unification, but the two concrete tweaks that carry the pitch, Multimodal AdaLN and feature pre-scaling, read more like sensible engineering choices than a new idea. Without a tiny swap experiment that turns AdaLN into plain layer norm or group norm and disables pre-scaling under the same tokens and backbone, it is difficult to credit the architecture rather than routine stabilization tricks.

2. **The evidence does not line up cleanly:** Some numbers in the text do not match the tables, and a few comparisons mix specialized models with unified ones, which makes it hard to judge where the method truly stands. Please reconcile the SEED and POPE figures, keep settings like prompts, decoding, and resolution identical across rows, and use ``competitive'' language when another model is ahead.

3. **Gains are entangled with the training recipe:** The three-stage schedule with teacher warmup, connector alignment, and unified tuning is likely doing a lot of the heavy lifting, yet there is no no-distillation control or compute-matched baseline to show how much comes from the architecture itself. A small control on a subset, even at reduced scale, would make the attribution more believable.

4. **Data hygiene and failure modes are not visible:** The training mix spans several large sources and possibly in-house data, yet there is no overlap or license audit against the evaluation suites, and the paper shows few concrete failure cases. A short data card plus a page of typical errors would go a long way toward showing the method is robust rather than cherry-picked.

**Questions:**

1. Report a tiny ablation on a small subset that swaps AdaLN → layer norm (or group norm) and turns pre-scaling off, with deltas on 1–2 key metrics.

2. Please reconcile the SEED and POPE numbers by providing a single aligned table for the main row and 2–3 baselines with the same prompts, decoding, resolution, and seeds, and indicate whether results are raw or rewritten.

3. Please include one minimal control to separate recipe effects from architecture, for example a short no-distillation run on a small subset or a compute-matched encoder-plus-adapter baseline, and summarize the observed delta.

---

### Official Review · Reviewer_Q7Cj · 2025-10-28

**Soundness:** 3
**Presentation:** 3
**Contribution:** 2
**Rating:** 4
**Confidence:** 4

**Summary:**

This paper addresses the challenge of building unified single-transformer models for multimodal understanding and generation, noting that existing methods often require prohibitively high training costs from "from-scratch" learning and exhibit performance limitations, especially in the video domain. It proposes an efficient training paradigm featuring a multimodal warmup strategy that leverages knowledge from pre-trained models. The paper also introduces feature pre-scaling and a multimodal AdaLN to mitigate cross-modal feature incompatibilities, thereby enhancing training stability and effectiveness. Building on these techniques, the paper introduces HaploOmni, a unified single transformer that supports both image and video understanding and video generation. Experiments presented by the authors show that HaploOmni achieves competitive or state-of-the-art performance across multiple image understanding benchmarks, outperforming prior unified models on video understanding and generation tasks.

**Strengths:**

* The paper clearly identifies and addresses two critical limitations of current unified multimodal models—prohibitively high training costs and performance gaps—which are both timely and impactful issues in the field.

* The proposed multimodal warmup strategy, leveraging knowledge from pre-trained models, combined with pre-scaling and multimodal AdaLN, effectively reduces training data and computational overhead while addressing cross-modal feature incompatibility. This leads to improved training stability, effectiveness, and accessibility.

* The paper introduces HaploOmni, which achieves competitive or state-of-the-art results on image understanding benchmarks and notably surpasses previous unified models in both video understanding and generation tasks.

**Weaknesses:**

* The performance evaluation for video understanding is insufficient. The comparisons are limited to EgoSchema and MVBench, omitting mainstream benchmarks such as MMBench-Video [1], VideoMME [2], and LongVideoBench [3]. In addition, the paper does not report results for image generation tasks against state-of-the-art models, which constitutes a core capability of unified multimodal understanding and generation models.

* Although the paper claims that HaploOmni is cost-efficient, it lacks a more thorough analysis, particularly in terms of training and inference overhead compared with other approaches.

* The advantages of HaploOmni in image understanding, as shown in Table 7, may largely stem from improvements in the LLM backbone. For instance, Show-o2 [4], which also uses Qwen2.5-7B as its LLM backbone, achieves better performance on image understanding benchmarks, suggesting that some gains may not be attributable to the proposed multimodal approach itself.

[1] Fang, Xinyu, et al. "Mmbench-video: A long-form multi-shot benchmark for holistic video understanding." Advances in Neural Information Processing Systems 37 (2024): 89098-89124.

[2] Fu, Chaoyou, et al. "Video-mme: The first-ever comprehensive evaluation benchmark of multi-modal llms in video analysis." Proceedings of the Computer Vision and Pattern Recognition Conference. 2025.

[3] Wu, Haoning, et al. "Longvideobench: A benchmark for long-context interleaved video-language understanding." Advances in Neural Information Processing Systems 37 (2024): 28828-28857.

[4] Xie, Jinheng, Zhenheng Yang, and Mike Zheng Shou. "Show-o2: Improved Native Unified Multimodal Models." arXiv preprint arXiv:2506.15564 (2025).

**Questions:**

refer to the weaknesses

---

### Official Review · Reviewer_i1zU · 2025-11-01

**Soundness:** 2
**Presentation:** 2
**Contribution:** 1
**Rating:** 2
**Confidence:** 3

**Summary:**

This paper addresses the challenge of efficient training for unified multimodal understanding and generation within a single transformer. It proposes HaploOmni, featuring a multimodal warmup strategy with prior knowledge, feature pre-scaling, and multimodal AdaLN to improve cross-modal compatibility. Experiments show that HaploOmni achieves competitive performance on image and video benchmarks with limited training cost, outperforming or matching advanced unified models.

**Strengths:**

The paper presents an efficient approach for unified understanding-and-generation models, accompanied by extensive experiments. The writing is clear and well-structured.

**Weaknesses:**

1. The proposed "Multimodal Warmup" and other architectural designs introduce significant complexity compared to simpler and more widely adopted paradigms for multimodal integration. Although the method achieves competitive overall performance, the necessity and generality of this complexity remain questionable, can it truly be transferred to other models?

2. The baselines compared in the paper are outdated. The authors should provide comparisons with recent state-of-the-art methods, such as Unitok, Bagel and other similar approaches cited in the paper, and discuss the advantages and disadvantages of each method.

3. Please provide a comparison of the data volumes used across the different methods. It is critical to clarify whether the performance improvements are attributable to the proposed methodological design or simply stem from scaling up the training data.



[1] Emerging Properties in Unified Multimodal Pretraining, 2025.05

**Questions:**

See the Weakness

---

### Official Review · Reviewer_Mvtz · 2025-11-05

**Soundness:** 3
**Presentation:** 2
**Contribution:** 3
**Rating:** 4
**Confidence:** 4

**Summary:**

This paper introduces an approach to unified multimodal models that progressively incorporates separated components into unified single-model frameworks. Specifically, a warm-up strategy is employed to utilize prior knowledge in visual encoders (e.g., CLIP) and visual generators (e.g., DiT). To address cross-modal compatibility challenges, feature pre-scaling and multimodal AdaLN are introduced to harmonize the features for understanding and generation tasks. With these priors and connecting designs,  the authors build the model named HaploOmni with significantly less training cost than existing single-transformer frameworks. In the experiment, HaploOmni was evaluated on both understanding and generation benchmarks for both images and videos.

**Strengths:**

1. The proposed method is quite resource-efficient, requiring only 20% generation data compared to ViLA-U. And the performance of HaploOmni beats existing models of similar scales by a clear margin.

2. The idea of utilizing understanding and generation priors in specialist models is intuitive and reduces the cost of "rebuilding wheels". The authors have explored several techniques to incorporate and harmonize the distilled features in a single transformer architecture.

3. The experimental verification is comprehensive, including both understanding and generation tasks for both images and videos.

**Weaknesses:**

1. **The motivation of Multimodal AdaLN:** This module is inherited from CogVideoX to facilitate visual generation. Why and how the design benefits the unification of understanding and generation is unclear. Can the authors provide some discussion and analysis on the design choice?

2. **Related Works:** It is advised to compare and discuss with more recent studies, such as :

[1] SynerGen-VL: Towards Synergistic Image Understanding and Generation with Vision Experts and Token Folding, Li et.al., CVPR 2025

[2] ILLUME: Illuminating Your LLMs to See, Draw, and Self-Enhance, Wang et.al., ICCV 2025

[3] Harmonizing Visual Representations for Unified Multimodal Understanding and Generation, Wu et.al., ICCV 2025

**Questions:**

1. Can the authors provide more details on the Feature Pre-scaling and relevant analyses (as discussed at L228-229)?

2. About the architectures of the Pre-decoder and Post-decoder, are these modules copied from CLIP and  CogVideoX? Or the same architecture as the base decoder?

3. Why is identity loss instead of NTP loss used in the warm-up stage for linguistic modelling?

---

### Note · Authors · 2025-11-24

I have read and agree with the venue's withdrawal policy on behalf of myself and my co-authors.